# Intensity of Physical Activity in Physical Education Classes and School Recesses and Its Associations with Body Mass Index and Global Fitness Score in Spanish Schoolchildren

José Francisco López-Gil [1,2,*], Iván Cavero-Redondo [2,3], Pedro Juan Tárraga-López [4], Edina Maria de Camargo [5], Irene Sequí-Domínguez [2], Juan Luis Yuste Lucas [6], Fernando Renato Cavichiolli [5] and Antonio García-Hermoso [7,8]

[1] Departamento de Actividad Física y Deporte, Facultad de Ciencias del Deporte, Universidad de Murcia (UM), 30720 San Javier, Spain
[2] Health and Social Research Center, Universidad de Castilla-La Mancha (UCLM), 16071 Cuenca, Spain; Ivan.Cavero@uclm.es (I.C.-R.); irene.SequiDominguez@uclm.es (I.S.-D.)
[3] Rehabilitation in Health Research Center (CIRES), Universidad de las Americas, Av. República 71, Santiago 72819, Chile
[4] Departamento de Ciencias Médicas, Facultad de Medicina, Universidad Castilla-La Mancha (UCLM), 02008 Albacete, Spain; pjtarraga@sescam.jccm.es
[5] Departamento de Educação Física, Universidade Federal do Paraná (UFPR), Curitiba 81531-980, Brazil; edinacamargo@gmail.com (E.M.d.C.); cavicca@hotmail.com (F.R.C.)
[6] Departamento de Expresión Plástica, Musical y Dinámica, Facultad de Educación, Universidad de Murcia (UM), 30100 Espinardo, Spain; jlyuste@um.es
[7] Navarrabiomed, Complejo Hospitalario de Navarra (CHN), Universidad Pública de Navarra (UPNA), IdiSNA, 31008 Pamplona, Spain; antonio.garciah@unavarra.es
[8] Escuela de Ciencias de la Actividad Física, El Deporte y la Salud, Universidad de Santiago de Chile (USACH), Santiago 71783, Chile
* Correspondence: josefranciscolopezgil@gmail.com

**Abstract:** Background: Examining the association between excess weight or physical fitness and intensity of physical activity (PA) during Physical Education (PE) classes or school recesses and in children could be of great interest and importance, especially for future intervention programs or public policies related to PA. The aim of this study was to explore the association between intensity of PA in PE classes or school recesses and excess weight or global physical fitness in a sample of Spanish schoolchildren. Methods: We performed a cross-sectional study in the Valle de Ricote (Region of Murcia, Spain). A final sample of 350 Spanish schoolchildren between 6 and 13 years of age was included from six different schools. Intensity of PA during PE classes and recesses was assessed by the Physical Activity Questionnaire for Older Children (PAQ-C). To assess physical fitness, the extended ALPHA fitness test battery was used. Body mass index (z-score) was calculated following the age- and sex-specific thresholds of the World Health Organization. Results: Both body mass index (z-score) and Global Fitness Score (z-score) were lower in schoolchildren engaging in high intensity of PA in both PE lessons and school ($p < 0.05$ for all). Lower odds of having excess weight were found in those who reported high intensity of PA in both PE classes and school recesses (OR = 0.54; CI 95%, 0.30–0.96). In these same participants, higher odds of being in the high Global Fitness Score tertile were found (OR = 1.96; CI 95%, 1.01–3.85). Conclusions: Our study showed that higher intensity of PA in PE classes and school recesses was associated with lower excess weight and higher global physical fitness.

**Keywords:** childhood obesity; cardiorespiratory fitness; muscular strength; school enrollment; children

## 1. Introduction

Excess weight prevalence, understood as the sum of overweight and obesity, has markedly increased in most high-income countries during the past three decades and,

according to the scarce data available in low- and middle-income countries, is growing at a worrying rate [1]. This is a worrisome situation since excess weight in childhood and adolescence is associated with harmful health outcomes throughout the life-course [2].

Moreover, childhood plays a crucial role in the acquisition of healthy habits. For instance, being physically active and having satisfactory levels of physical fitness are crucial for present and future health [3]. Physical fitness level (e.g., cardiorespiratory fitness, muscular fitness) during childhood and adolescence is associated with more appropriate health markers later in life [4,5]. Nevertheless, there has been a substantial decline in some components of physical fitness (e.g., cardiorespiratory fitness), suggesting a significant decline in young people health [6,7].

According to the World Health Organization, children and adolescents should engage in an average of 60 min of moderate-to vigorous physical activity (MVPA) daily, due to its demonstrated benefits for both weight status and physical fitness, among others [8], However, in Spain, the young population engage in low levels of PA intensity [9], as well as in the Region of Murcia [10,11]. In this sense, the school setting, and physical education (PE) teachers in particular, must exert their role in public health by following a more comprehensive school PA program [12], including quality PE classes [13,14]. PE classes and school recesses are suitable school settings to promote physical activity (PA), since most children attend school and could be targeted [15]. In this regard, one systematic review concluded that school contributes to more than 40% of the total MVPA. Similarly, Gao et al. [16] found that PE was more effective for engaging in MVPA than other segments during the school day. Moreover, a metanalysis by Parrish et al. [17] concluded that school recess interventions show promise for increasing MVPA.

Despite the above-mentioned health effects and the feasibility of engaging in PA within the school context, some studies have reported low levels of PA during both PE classes and school recesses. For instance, a meta-analysis published by Truelove et al. [18] showed that, on average, children engaged in MVPA for 33.0% of their PE classes and were sedentary for 35.9% of their PE classes. Tercedor et al. [19] also found low levels of MVPA during school recess are in children, a low percentage of children meet the MVPA recommendation during school recess, and overweight/obese children show still lower values in MVPA. Similarly, some meta-analyses have found that the proportion of PE lesson time during which young population are engaged in MVPA is typically less than 50% of the recommendation suggested by the World Health Organization (WHO) [20,21].

However, only a few studies have analyzed the association between excess weight and physical fitness in relation to PA at school setting. For instance, a systematic review by Gray et al. [22] found several factors associated with lower rates of obesity or obesity trajectory, such as increased minutes of recess, meeting with recommended recess and PE time, among others. Regarding physical fitness, Chen et al. [23] found a higher performance in four different physical fitness tests in those schoolchildren physically active during PE and in school recess. Furthermore, Calahorro-Cañada et al. [24], showed that young people who were sedentary for more than 15 min during recess time presented lower odds of cardiorespiratory fitness.

Examining the association between PA intensity during PE classes or school recesses and excess weight or global physical fitness and in children could be of great interest and importance, especially for future intervention programs or public policies related to PA. This relevance lies in three main reasons which reinforces the need of this study: first, that children spend a large part of their time in-and-around the school environment; second, the crucial role of maintaining an adequate weight status and a high physical fitness in promoting and maintaining health; and third, the low prevalence found of children meeting with the PA recommendation in Spain and in the Region of Murcia. Thus, our hypothesis is that schoolchildren who apply a higher PA intensity in PE classes and during recesses would have lower odds of having excess weight and higher odds of having high global physical fitness compared to those who apply a lower intensity of PA. Thus, the aim of

this study was to explore the association between intensity of PA in PE classes or school recesses and excess weight or global physical fitness in a sample of Spanish schoolchildren.

## 2. Material and Methods

### 2.1. Design and Participants

We performed a cross-sectional study in the Valle de Ricote. Five different municipalities (Archena, Villanueva del Río Segura, Ricote, Ulea and Ojós) compound the Valle de Ricote, located in the Region of Murcia (Spain). All schools in these different municipalities were asked to participate in this study. Participants were enrolled by convenience sampling from six different schools (private with public funds and public). Although we used this sampling technique, participation was requested from all schoolchildren in the available schools. Finally, a sample of 350 Spanish schoolchildren between 6 and 13 years of age was included in this study.

Regarding the participation in this study, the parents/legal guardians of the participants received a written consent form that they were required to sign to allow their children to participate. Furthermore, a brief explanation was provided to these parents/legal guardians and their children about the purposes of this study, and the tests and questionnaires that would be conducted. Concerning the inclusion criteria, only schoolchildren aged 6 to 13 years with parents/legal guardians who signed the informed consent were enrolled. By contrast, the exclusion criteria were as follows: (a) schoolchildren who were exempt from the subject of PE at school, since the anthropometric measurements, physical tests and the fulfilment of the questionnaires were performed during PE classes; (b) schoolchildren with some type of dysfunction that limited the engagement in PA (i.e., some disease or motor problem); and (c) participants under some type of pharmacological treatment.

The Bioethics Committee of the University of Murcia approved the current study (ID 2218/2018). Furthermore, it was performed following the Helsinki Declaration and with full respect of the human rights of the included schoolchildren.

### 2.2. Procedures

#### 2.2.1. Anthropometric Data

The body weight of the schoolchildren was determined with an electronic scale, accurate to 0.1 kg (Tanita BC-545, Tokyo, Japan), and their height was measured with a portable measuring rod accurate to 0.1 cm (Leicester Tanita HR 001, Tokyo, Japan). The body mass index (BMI) of the schoolchildren was computed by the ratio of their body weight (kg) to their squared height (m$^2$). Further, BMI z-score (zBMI) was calculated following the age- and sex-specific thresholds of the WHO [25] and, consequently, the proportion of excess weight (overweight and obesity) of schoolchildren was determined.

#### 2.2.2. Global Fitness Score

To assess physical fitness, the extended ALPHA fitness test battery was used, which had already been explained and validated in young people [26]. Data on muscular strength (both lower body and upper body), cardiorespiratory fitness, and speed-agility were collected using Standing Broad Jump, Handgrip Strength Test, 20 m Shuttle Run Test, and 4 × 10 Shuttle Run Test, respectively. Participants completed all the tests during their PE classes. The individual score of each test was converted into sex- and age-specific standardized values (z-scores). Since a longer time on the 4 × 10 Shuttle Run Test reflects lower speed agility, the z-score of this test was inverted. A global fitness score (GFS) was computed as the mean of all the z-scores values. Previous studies have applied this approach to determine the overall global fitness [27]. A higher z-score in the GFS denotes higher physical fitness performance. In addition, GFS was divided into tertiles to obtain three balanced groups: (1) low zGFS, (2) medium zGFS, and (3) high zGFS. For further analyses, we collapsed these categories into: (1) low/medium GFS tertile and (2) high GFS tertile.

### 2.2.3. Intensity of Physical Activity in Physical Education Classes and Recesses

The schoolchildren filled the Physical Activity Questionnaire for Older Children (PAQ-C) [28], which was previously validated and adjusted to Spanish children [29]. The PAQ-C was created and validated to compute the levels of MVPA in children aged 8 to 14 years. Therefore, in the case of schoolchildren aged 6–7, we followed the suggestions by Bervoets et al. [30], who recommended encouraging parents to support their children in reading and to complete the questionnaire, provided that they did not give any guidance in answering the questions. Thus, intensity of PA in PE classes was established from the following question "In the last 7 days, during PE classes, how often were you very active during classes: playing intensely, running, jumping, throwing?". The possible responses were: 1—Almost never; 2—Sometimes; 3—Often; 4—Always. Additionally, we created two different groups: (1) low-intensity PE (almost never, sometimes, and often) and (2) high-intensity PE (always). Intensity at recess was determined by the following items. "In the last 7 days, what did you do most of the time at recess?" Possible responses were 1—Sitting, 2—Walking, 3—Playing a little, 4—Playing a lot, 5—Playing intensely. For subsequent analyses, we collapsed these categories into: low-intensity recess (i.e., sitting, walking, playing a little, and playing a lot) and high-intensity recess (i.e., playing intensely).

### 2.2.4. Covariates

Sex and age were self-reported by the schoolchildren. The type of schooling was categorized into two groups: private with public funds and public. Area of residence was divided into (1) rural ($\leq$5000 inhabitants) or urban (>5000 inhabitants) [31]. Schoolchildren who met at least one of the following assumptions were considered to be immigrants: (a) immigrant parents, (b) at least one of their parents comes from another country, or (c) were born outside Spain. Furthermore, participants were asked about the number of hours that they usually spend as daily to screen time (TV or play video games) and were asked about weekly hours of sport activities were asked using the Krece Plus Short Test, which was validated by the enKid Study for children and preadolescents aged 4 to 14 years [32].

### 2.3. Statistical Analyses

Descriptive data are presented as means and standard deviation for continuous variables and frequencies and percentages for categorical variables. Chi-squared was applied to test for differences between categorical variables. An analysis of covariance (ANCOVA) was used to estimate differences between mean values of zBMI and zGFS across PA intensity in PE classes (high or low), and PA intensity in school recess (high or low) groups. Furthermore, ANCOVA with post hoc pairwise comparisons using Bonferroni was used to test for differences between mean values of zBMI and zGFS across combined intensity in PE classes and school recesses categories, using four groups for each combination: (1) "High-intensity PE and High-intensity recess" (high intensity of PA in PE, and high intensity of PA in school recess); (2) "High-intensity PE and Low-intensity recess" (high intensity of PA in PE, and low intensity of PA in school recess); (3) "Low-intensity PE and High-intensity recess" (low intensity of PA in PE, and high intensity of PA in school recess); (4) "Low-intensity PE and Low-intensity recess" (high intensity of PA in PE, and high intensity of PA in school recess). Since we did not find significant interactions between sex and different intensities of PA in PE or school recesses in relation to zBMI (PE: $p = 0.839$; and school recesses: $p = 0.593$) and zGFS (PE: $p = 0.593$; and school recesses: $p = 0.457$) in the initial analyses, both girls and boys were analyzed together for greater statistical power. Furthermore, binary logistic regression analyses were carried out to verify the association between excess weight (no excess weight versus excess weight) and GFS tertiles (low/medium GFS tertiles versus high GFS tertile) and the different PA intensity in PE classes, school recesses, and both settings. All analyses were adjusted for age, sex, type of schooling, area of residence, daily screen time, and weekly sport activities. Data analy-

ses were performed using the Statistical Package for Social Sciences (v25.0) software. A *p*-value < 0.05 was considered as statistically significant.

## 3. Results

The characteristics of the study participants are shown in Table 1. The mean age was 8.7 years old (SD = 1.8). The proportion of participants with excess weight was 42.3%.

**Table 1.** Descriptive data of the Spanish schoolchildren analyzed (N = 350).

| Variables | M (SD)/n (%) | CI 95% |
|---|---|---|
| Age (years) | 8.7 (1.8) | 8.5–8.9 |
| Sex, (%, boys) | 193 (55.1) | 49.9–60.4 |
| Type of schooling (%, public) | 234 (66.9) | 61.9–71.8 |
| Area of residence (%, urban) | 271 (77.4) | 73.0–81.8 |
| Immigrant status (%, native) | 184 (81.1) | 47.3–57.8 |
| Weekly sport activities | | |
| None (%) | 31 (8.9) | 5.9–11.8 |
| 1 h (%) | 53 (15.1) | 11.4–18.9 |
| 2 h (%) | 97 (27.7) | 23.0–32.4 |
| 3 h (%) | 52 (14.9) | 11.1–18.6 |
| 4 h (%) | 32 (9.1) | 6.1–12.2 |
| 5 or more h (%) | 85 (24.3) | 19.8–28.8 |
| Daily screen time | | |
| None (%) | 26 (7.4) | 4.7–10.2 |
| 1 h (%) | 143 (40.9) | 35.7–46.0 |
| 2 h (%) | 126 (36.0) | 31.0–41.0 |
| 3 h (%) | 38 (10.9) | 7.6–14.1 |
| 4 h (%) | 11 (3.1) | 1.3–5.0 |
| 5 or more h | 6 (1.7) | 0.4–3.1 |
| Anthropometric data | | |
| Weight (kg) | 35.9 (10.9) | 34.8–37.0 |
| Height (cm) | 135.9 (11.8) | 134.7–137.1 |
| BMI (z-score) [a] | 1.14 (1.24) | 1.01–1.27 |
| Excess weight [a] (%) | 148 (42.3) | 37.1–47.5 |
| Physical fitness | | |
| 20 m Shuttle Run Test (stages) | 2.7 (1.5) | 2.5–2.9 |
| Cardiorespiratory fitness (mL/kg/min) | 45.0 (4.2) | 44.8–45.2 |
| Handgrip strength (kg) | 13.1 (4.0) | 12.7–13.5 |
| Handgrip strength/body weight | 0.37 (0.08) | 0.36–0.38 |
| Standing broad jump (cm) | 113.8 (25.0) | 111.2–116.4 |
| 4 × 10 Shuttle Run Test (s) | 13.74 (1.43) | 13.59–13.89 |

Data expressed as numbers (percentages) or mean (standard deviation), and 95% confident intervals. [a] Excess weight (overweight and obesity) determined by the World Health Organization criteria [25]. BMI, body mass index.

Figure 1 shows the proportion of schoolchildren performing in different intensities of PA during PE classes and school recesses. Overall, 49.1% of the schoolchildren reported to be very active "always". Furthermore, the 26.6% of the analyzed sample declared "play intensely" during the school recesses. Moreover, the proportion of schoolchildren with high-intensity during both PE classes and school recesses was 38.9%.

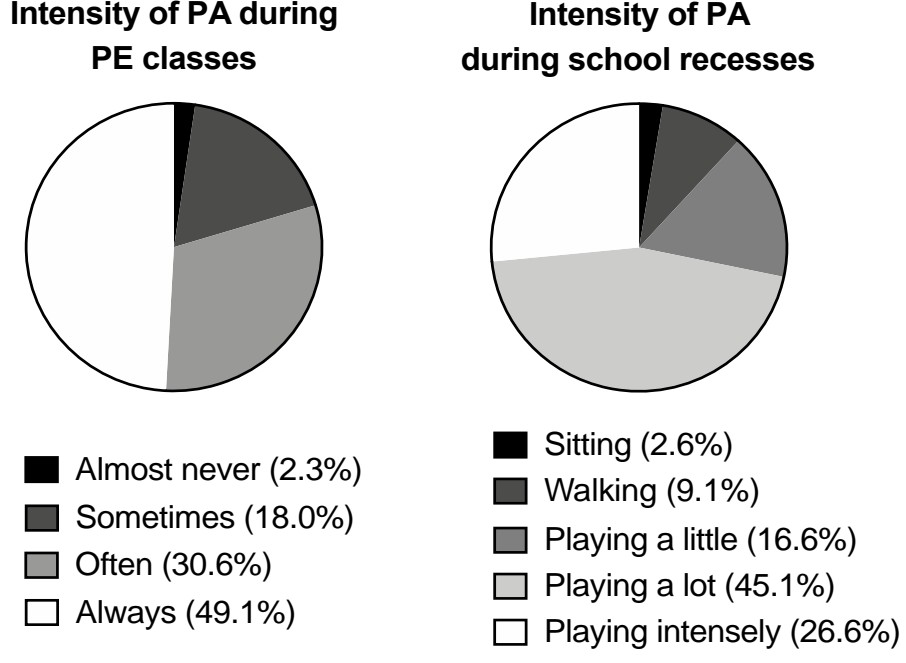

**Intensity of PA during PE classes**

- ■ Almost never (2.3%)
- ▨ Sometimes (18.0%)
- ▨ Often (30.6%)
- ☐ Always (49.1%)

**Intensity of PA during school recesses**

- ■ Sitting (2.6%)
- ▨ Walking (9.1%)
- ▨ Playing a little (16.6%)
- ▨ Playing a lot (45.1%)
- ☐ Playing intensely (26.6%)

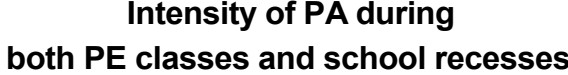

**Intensity of PA during both PE classes and school recesses**

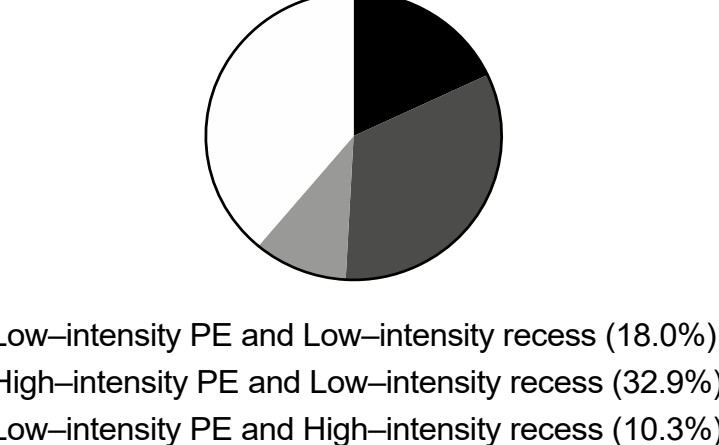

- ■ Low–intensity PE and Low–intensity recess (18.0%)
- ▨ High–intensity PE and Low–intensity recess (32.9%)
- ▨ Low–intensity PE and High–intensity recess (10.3%)
- ☐ High–intensity PE and High–intensity recess (38.9%)

**Figure 1.** Proportion of schoolchildren reporting different intensities of physical activity according to Physical Education classes and school recesses, individually and in combination. PA, physical activity; PE, physical education.

Figure 2 shows the mean differences between zBMI and zGFS according to PA intensity during PE classes and school recesses. Thus, zBMI was lower in those who reported higher intensity of PA in PE lesson, school recesses and both settings ($p < 0.05$ for all). Conversely, zGFS was higher in schoolchildren who engaged in higher intensity of PA in PE classes, school recesses, and both settings ($p < 0.05$ for all).

## Body Mass Index (z-score)

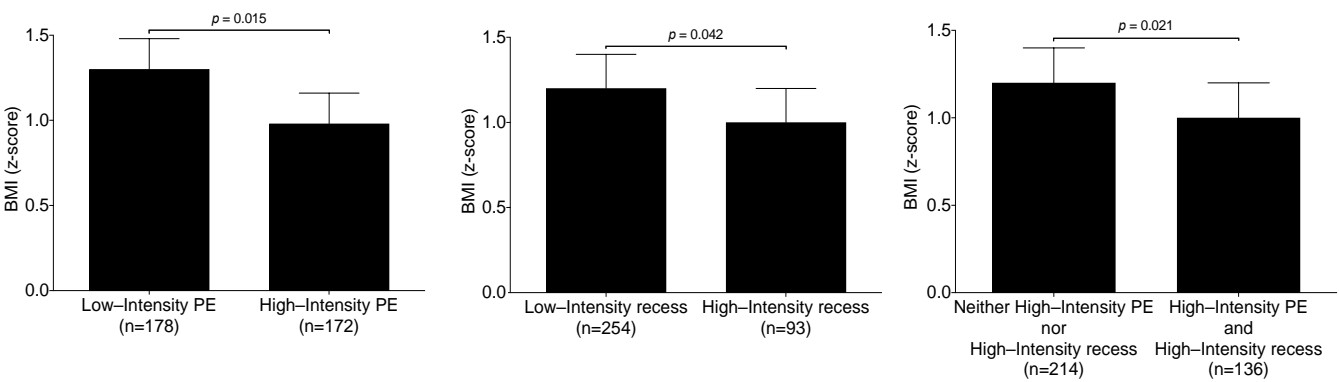

## Global Physical Fitness (z-score)

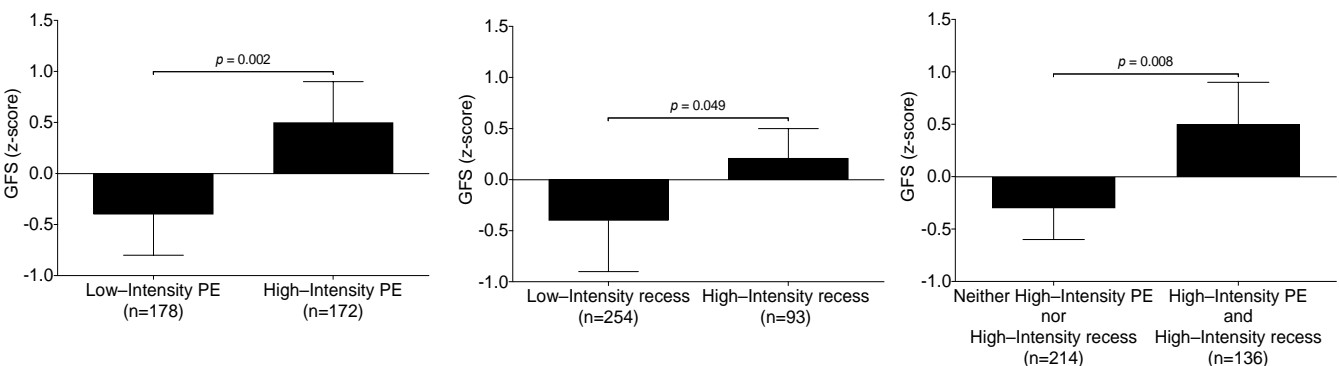

**Figure 2.** Mean differences in body mass index (z-score) and global physical fitness (z-score) in relation to physical activity intensity during Physical Education classes and school recesses. Analyses were adjusted by sex, age, type of schooling, area of residence, immigrant status, daily screen time, and weekly sport activities. BMI, body mass index; GFS, Global Fitness Score; PE, physical education.

Figure 3 indicates the odds ratio of having excess weight or being in the high GFS tertile in relation to PA intensity during PE classes, school recesses, or both settings. A lower odd of having excess weight was found in those engaging in high intensity of PA in PE classes (OR = 0.55; CI 95%, 0.37–0.84), and in both PE lesson and school recesses (OR = 0.54; CI 95%, 0.30–0.96). Conversely, higher odds of being in the high GFS tertile were found in schoolchildren engaging in high intensity of PA in PE classes (OR = 1.96; CI 95%, 1.25–3.08), as well as in both PE lessons and school recesses (OR = 1.96; CI 95%, 1.01–3.85).

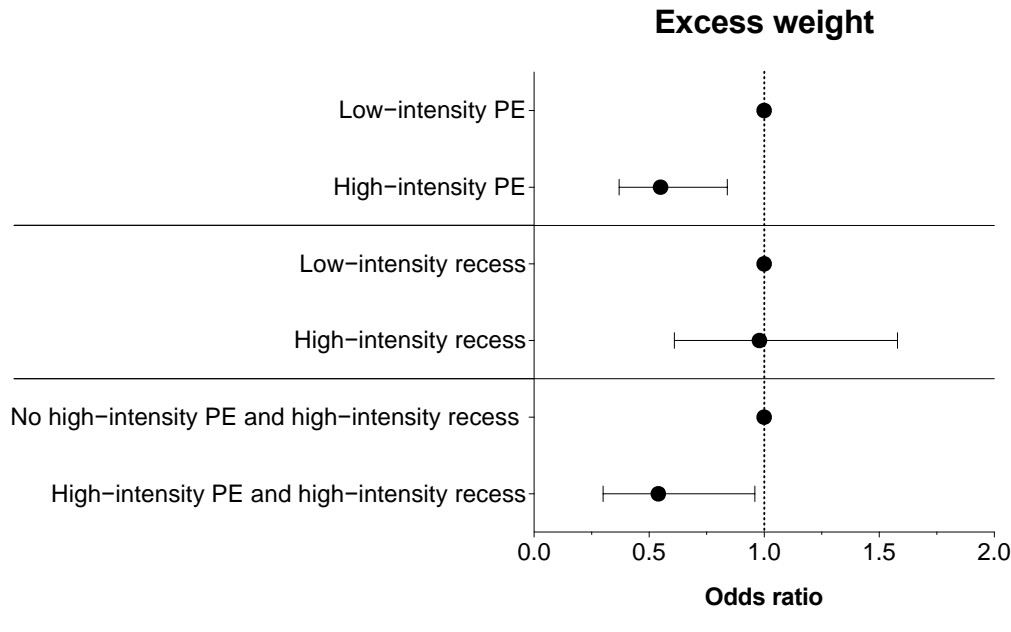

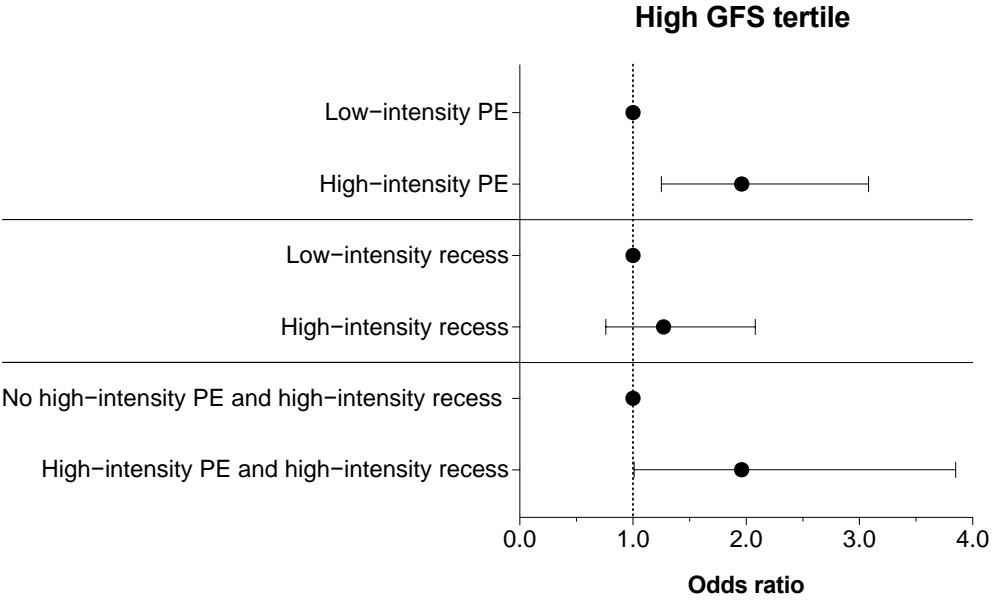

**Figure 3.** Association between physical activity intensity during Physical Education classes and school recesses and excess weight and global physical fitness according. Analyses were adjusted by sex, age, type of schooling, area of residence, immigrant status, daily screen time, and weekly sport activities. BMI, body mass index; GFS, Global Fitness Score; PE, physical education.

## 4. Discussion

The current study found that PA intensity in PE classes and in school recesses (individually and combined) was associated with excess of weight and physical fitness among our sample of Spanish schoolchildren. In this study, only a few schoolchildren reported high intensity of PA in both PE classes and school recesses. The lack of studies using this methodology to assess the intensity of PA in school context makes discussion difficult. However, our finding is in line with previous studies among schoolchildren [33,34]. For instance, Ronghe et al. [33] found that 31% of the Indian schoolchildren analyzed were always active during PE classes, and 38% ran and played intensely. Similarly, Wong et al. [34] indicated that only 38.7% were "always" active during PE classes, in their study among Malaysian schoolchildren. Conversely, another study among Colombian schoolchildren

found that 30% of schoolchildren remained seated during recess. These same authors shown that 8% of participants ran and played intensely [35]. In addition, one study by Zuñiga-Galaviz et al. [36] found that the less than the 3% of Mexican schoolchildren analyzed spent their time in sedentary behaviors during recess, much lower than the results reported by Dessing et al. [37], who found higher than 70% in Dutch schoolchildren. In Spain, Llorente-Cantarero et al. [38] observed that 86.7% of the students were active during recess, compared to 13.3% who were sedentary in schoolchildren in the Region of Murcia. The low levels of PA intensity found in our study also seem to be in line with the low levels of MVPA reported in Spain [9], as well as in the Region of Murcia [10,11]. However, given the scarce information found in the literature, as well as the different methodologies applied to determine the intensity of PA during PE classes or school recesses (different PA questionnaires, GPS, accelerometers, etc.), caution is required when interpreting these results.

Regarding BMI, our results showed lower zBMI in those who reported higher intensity of PA during PE classes, school recesses, and both settings. In addition, we found a lower odd of having excess weight in participants who reported higher intensity of PA during PE classes, and both PE classes and school recesses. Our results are in line with a recent study by Burns et al. [39], which showed that lower zBMI was significantly linked to a higher percentage of steps (e.g., higher PA level) accrued during both PE classes and school recesses. Similarly, Fenech et al. [40] found that engaging in more than 50% of PE classes in MVPA was associated with reductions in zBMI among Maltese children. Additionally, Thalken et al. [41] found that the presence of school policies related to recess access significantly predicted lower BMI in schoolchildren, possibly because much of the PA that takes place during the school day occurs at school recess [42]. Supporting this idea, children seem to engage more minutes in MVPA during the most of school day segments, which has been associated with lower zBMI [43]. Conversely, another study showed no significant association between excess weight and PA school recess among Brazilian adolescents [44]. However, this study was performed in older participants and in a different country to ours study, which could explain (at least partly) the mismatch. In relation to global physical fitness, we found higher zGFS in those schoolchildren engaged in high intensity of PA in PE classes, school recesses, and both settings. Similarly, a greater odd of being in the high GFS tertile in those who reported high intensity of PA in PE classes, and in both PE and recesses. This finding is in line with one study by Chen et al. [23] who found higher performance in four different physical fitness tests in those schoolchildren who were physically active during PE classes and school recesses. Furthermore, Calahorro-Cañada et al. [24], found a higher cardiorespiratory fitness in young people who were sedentary for more than 15 min during recess time. Likewise, Coledam et al. [45] found that higher physical fitness (e.g., cardiorespiratory fitness, muscular strength) was associated with being active during PE classes among Brazilian schoolchildren. Conversely, Cheung et al. [46] found that physical fitness was not related to during-school PA in Georgian schoolchildren. This discrepancy could be explained by the different methodology applied (i.e., FitnessGram test) or by the fact that the information was reported by the teachers and not by the schoolchildren. There are some potential reasons justifying our results. First, children who spend less time on PA during both PE classes and school recesses spend more time in sedentary behaviors (characterized by a low energy expenditure), which are associated with undesirable health outcomes (e.g., excess weight, low physical fitness) [8]. Moreover, replacing MVPA with any other movement behavior (e.g., sedentary behavior) predicted higher adiposity and lower physical fitness (e.g., cardiorespiratory fitness) in children [47]. Second, a higher PA intensity demands a greater energy expenditure [48], and, consequently, schoolchildren who engage in higher PA intensity expend more energy expenditure than their counterparts [49]. In this sense, Huddleston et al. [50] found that PE classes and recesses are the school settings that generate the highest energy expenditure. Third, higher PA intensity is also associated with health benefits in the young population (e.g., normal weight, higher physical fitness) [8,51], which could also be applicable in the

school context. Fourth, another possible explanation could lie in the relationship between the PA intensity and a healthier diet. Eating healthy foods is associated with higher odds of being physically active (i.e., more MVPA) and less sedentary in children, compared to their counterparts consuming less healthy meals [52]. Supporting this, a healthy diet is associated with a more adequate weight status [53] and high physical fitness [54].

This study is not without limitations. First, the cross-sectional precludes establishing cause-effect relationships, and reverse causality may occur if children with excess weight or low GFS engage in lower PA intensity during PE classes or school recesses (or both), and vice versa. Second, although we used validated PA questionnaires, we did not use more objective measure (e.g., accelerometers), which would have provided a more accurate estimation of PA level. Third, BMI is a limited measure of the extent and distribution of body fat, but it has the great advantage of having consistent and comparable data in many population-based surveys, especially compared with other methods, such as dual-energy x-ray absorptiometry (DEXA), which more complex and expensive [35]. Moreover, a systematic review concluded that DEXA and BMI were highly correlated [55] for determining body fat. Fourth, the failure to consider the developmental status of the participants could have introduced a confusing effect. Finally, we have not taken into account if children had access to equipment in the school recess, since those can help increase PA [56] and provide opportunities for children to select physical activities of their choice. Despite these limitations, this study includes some strengths. For instance, we used objective measures to determine excess weight and physical fitness (e.g., BMI, ALPHA fitness test battery [26]) in the sample of schoolchildren analyzed. Furthermore, since only a few studies have analyzed the association between PA at school setting and excess weight or physical fitness, we contribute to increase the scientific knowledge of this understudied topic.

## 5. Conclusions

Our study showed that intensity of PA in PE classes and school recesses were related to body mass index and global physical fitness. The current data highlight the importance of the promotion of higher intensity in a school setting, mainly at PE classes and school recesses, to favor possible improvement in the health status of schoolchildren.

**Author Contributions:** Conceptualization, J.F.L.-G.; methodology, J.F.L.-G.; software, J.F.L.-G.; validation, J.F.L.-G.; for-mal analysis, J.F.L.-G.; investigation, J.F.L.-G.; resources, J.F.L.-G.; data curation, J.F.L.-G.; writing—original draft preparation; I.C.-R., P.J.T.-L., E.M.d.C., I.S.-D., J.L.Y.L., F.R.C. and A.G.-H. contributed to the revision of the manuscript. All authors have read and agreed to the published version of the manuscript.

**Funding:** This study was partially financed by the Coordenação de Aperfeiçoamento de Pessoal de Nível Superior—Brasil (CAPES)—Finance Code 001.

**Institutional Review Board Statement:** The study was conducted according to the guidelines of the Declaration of Helsinki and was approved by the Bioethics Committee of the University of Murcia (ID 2218/2018).

**Informed Consent Statement:** Informed consent was obtained from all subjects involved in the study.

**Data Availability Statement:** The data presented in this study are available on request from the corresponding author. The data are not publicly available because they belong to minors.

**Acknowledgments:** The authors would like to express their gratitude to the Universidad de Murcia and the Ayuntamiento de Archena, as well as the participation of all the children, parents/legal guardians, physical education teachers, schools, and staff implicated, and wish to thank them for the appreciated information provided. J.F.L.-G. is a Postdoctoral Fellow (Universidad de Castilla-La Mancha—ID 2021-UNIVERS-10414). A.G.-H. is a Miguel Servet Fellow (Instituto de Salud Carlos III—CP18/0150).

**Conflicts of Interest:** The authors declare no conflict of interest.

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
