# Peer review of "Intensity of Physical Activity in Physical Education Classes and School Recesses and Its Associations with Body Mass Index and Global Fitness Score in Spanish Schoolchildren"

_applsci, doi:10.3390/app112311337_

Round 1

Reviewer 1 Report

Thank you for the material sent for evaluation. The limitations indicated by the authors are very important and seem to lower the value of the entire study. In addition, there are logical errors in the test (in abstact the authors provide: "High-intensity of PA in both PE lesson and school recesses was related to higher body mass index (...)" while in the summary: "Our study showed that higher intensity of PA in PE lessons and school recesses was associated with lower excess weight (...) - requires verification) and editorial: the last 3 paragraphs of the discussion are a repetition of the higher paragraphs. The whole thing still needs to be considered. 

The value of the study is reduced by the use of subjective methods. Data were collected for objective - fitness test results but were not discussed in the study. It is worth re-examining the collected material and considering remodeling the study. 

Reviewer 2 Report

Review of the article Intensity of Physical Activity in Physical Education Classes and School Recesses and its Associations with Body Mass Index and Global Fitness Score in Spanish Schoolchildren

This article deals with the relationship between PA intensity in PE lessons or school breaks and overweight or exercise in a sample of Spanish school children.

Abstract

Correct, succinctly described. It takes into account all the important points.

Introduction

You can see a lot of work by the authors in this section. The authors list many examples of other authors (other studies) who analyzed PA in school. Importantly, these are publications from various countries. It is worth adding that most of the cited publications are from the last few years.

Material and Methods

Design and participants In this section, they describe each part of the procedures well, including but not limited to selection of the test sample. In the section "Intensity of physical activity in Physical Education classes and recesses" the authors write well that they used the PAQ-C questionnaire, and for children aged 6-7 years they asked their parents for help. Correct application of statistical analyzes

Results

However, analyzing tables no. 1, it can be concluded that reporting by children and their parents information, e.g. Screen times are rather imperfect. In future research, it is worth using e.g. accelerometers (and citing more publications, e.g. on screen time and MVPA - e.g. you can find in: Herbert, J .; MatÅ‚osz, P .; Lenik, J .; Szybisty, A .; Baran, J .; Przednowek, K .; WyszyÅ„ska, J. Objectively Assessed Physical Activity of Preschool-Aged Children from Urban Areas. Int. J. Environ. Res. Public Health 2020, 17, 1375. https://doi.org /10.3390/ijerph17041375) or: Dahlgren A, Sjöblom L, Eke H, Bonn SE, Trolle Lagerros Y. Screen time and physical activity in children and adolescents aged 10-15 years. PLoS One. 2021 Jul 9; 16 (7): e0254255. doi: 10.1371 / journal.pone.0254255. PMID: 34242329; PMCID: PMC8270173. Moreover, very good analysis

Discussion

First, the text in the article is repeated twice in its entirety! The authors write about:… a higher percentage of steps accrued during both PE classes and school recesses, was significantly linked to lower zBMI… - but that was not the aim of the article. Also, school recess was not analyzed in the research results! The authors describe limitations well.

References

Very good and currently selected literature. 

Round 2

Reviewer 1 Report

Thank you for the changes and corrections made. I encourage you to consider preparing material based on objective and not only subjective measurements.